# Sex Differences in Short- and Long-Term Survival after Acute Type A Aortic Dissection

**DOI:** 10.3390/medicina60030443

**Published:** 2024-03-07

**Authors:** Philipp Pfeiffer, Lena Brendel, Romina Maria Rösch, Chris Probst, Ahmed Ghazy, Edoardo Zancanaro, Hazem El Beyrouti, Hendrik Treede, Daniel-Sebastian Dohle

**Affiliations:** 1Department of Cardiovascular Surgery, University Medical Center Mainz, 55131 Mainz, Germany; lena.brendel@med.uni-heidelberg.de (L.B.); romina.roesch@med.uni-heidelberg.de (R.M.R.); chris.probst@unimedizin-mainz.de (C.P.); ahmed.ghazy@unimedizin-mainz.de (A.G.); e.zancanaro96@gmail.com (E.Z.); hbeyrouti@gmail.com (H.E.B.); hendrik.treede@unimedizin-mainz.de (H.T.); daniel-sebastian.dohle@unimedizin-mainz.de (D.-S.D.); 2Department of Thoracic Surgery, Thoraxklinik Heidelberg, Heidelberg University Hospital, 69126 Heidelberg, Germany

**Keywords:** acute type A aortic dissection, sex differences, long-term survival, cardiac surgery, aortic surgery

## Abstract

*Background and Objectives*: Acute type A aortic dissection (AAD) is a life-threatening disease. No differences between men and women have been made in the treatment of AAD so far and knowledge about sex differences regarding long-term outcomes is limited. *Materials and Methods*: Between 01/2004 and 12/2021, 874 patients were operated on for AAD, including 313 (35.8%) women and 561 (64.2%) men. Clinical and surgical records, including long-term follow-up information, were obtained and analyzed retrospectively. To account for differences in the outcome determined by different preoperative life expectancies, a subgroup analysis for a set of patients matched according to their remaining life expectancy was performed. *Results*: At the time of AAD, women were older than men (69.1 ± 13.0 vs. 61.8 ± 13.3 years, *p* < 0.001) and had a shorter remaining statistical life expectancy (18.6 ± 10.8 vs. 21.4 ± 10.4 years, *p* < 0.001). Significantly more DeBakey type II AAD was found in women (37.1% vs. 25.7%, *p* < 0.001). Comorbidities and preoperative status at the time of presentation were similar in women and men. More hemiarch procedures (63.3% vs. 52.0%, *p* < 0.001) and less arch replacements (8.6% vs. 16.6%, *p* < 0.001) were performed in women, resulting in shorter cross-clamp times for women (92 ± 39 vs. 102 ± 49 min, *p* < 0.001). The in-hospital mortality was similar in women and men (11.5% vs. 12.7%, *p* = 0.618). Long-term survival was significantly shorter in women compared to men (9.8 [8.1–11.5] vs. 15.1 [11.9–18.4] years, *p* = 0.011). A matched subgroup analysis revealed that when comparing groups with a similar remaining life expectancy, the long-term survival showed no significant differences between women and men (9.8 [7.9–11.6] vs. 12.4 [10.1–14.7] years, *p* = 0.487). *Conclusions*: There are sex differences in AAD, with DeBakey type II dissection being more frequent in women. The seemingly worse long-term outcome can mostly be attributed to the shorter remaining statistical life expectancy at the time of presentation.

## 1. Introduction

Acute type A aortic dissection (AAD) is a life-threatening disease requiring emergent surgical repair. A high mortality has been observed for untreated patients, especially during the first 48 h of symptom onset [1]. The incidence in the general population is difficult to estimate due to out-of-hospital deaths prior to diagnosis. In Germany, the incidence has been described at about 5 per 100,000 persons per year, although autopsy studies suggest higher incidences [2,3] of up to 12 per 100,000 persons per year. The treatment of choice is an emergent surgical treatment of the ascending aorta and possibly the aortic arch, depending on the extension of the aortic dissection as well as other factors. Only 39% of all patients are diagnosed within the first 24 h of symptom onset and the female sex is one of the variables associated with a late diagnosis [4]. So far, no differences between men and women have been made in the treatment of acute type A aortic dissections. While some studies suggest a higher survival in the short-term for male patients, most do not account for the overall higher age of female patients at the time of the surgery nor for the different life expectancies of men and women [5,6,7]. Further knowledge about sex differences, particularly in long-term outcomes, is limited [8,9,10]. In cardiothoracic surgery, the impact of sex on outcomes has received some attention and a higher mortality rate for female patients undergoing isolated coronary artery bypass grafting in a meta-analysis of short-term, mid-term and long-term outcomes has been found [11].

This study aims to analyze the demographics, comorbidities, operative details and postoperative course and to compare the short- and long-term outcomes with respect to the remaining statistical life expectancy at presentation.

## 2. Materials and Methods

### 2.1. Patient Selection

All patients who were surgically treated for an acute type A aortic dissection between January 2004 and December 2021 were identified in our institutional database using the ICD-10-GM classification and included in this analysis. Patients who died prior to the initiation of surgery, or where the diagnosis of AAD was ultimately not confirmed, were excluded.

Demographics, comorbidities, operative details, postoperative course, and short-term and long-term outcomes were retrospectively analyzed with data obtained from the institutional database.

The present study was conducted in accordance with the Declaration of Helsinki and approved by the institutional ethics committee of the medical association of the state of Rheinland-Pfalz (2018-13574-Epidemiologie). Informed patient consent was waived due to the retrospective design of this study.

### 2.2. Procedural Details

The surgeries in this study were performed in a single center. Although some procedural details varied over the course of 18 years between different surgeons, the common standards are described below. The AAD was diagnosed by computed tomography, transoesophageal or transthoracic echocardiography, or coronary angiography. All patients diagnosed with AAD were immediately admitted to the operating room. At least two arterial pressure lines, usually the right radial artery and left femoral artery, a central venous line, and a sheath introducer or Shaldon’s catheter were established.

Cerebral perfusion was routinely monitored by bilateral cerebral oximetry (INVOS Somanetics, Troy, MI, USA) and body core temperature was measured vesically. Transesophageal echocardiography was routinely performed. Several arterial cannulation sites have been used, but more recently, the cannulation strategy at our center has changed to primary cannulation of the subclavian artery in stable patients prior to sternotomy. After median sternotomy and opening of the pericardium, the aorta and aortic arch were prepared for cannulation or replacement. Heparin was administered and cardiopulmonary bypass (CPB) was established. Patients were then cooled to a designated temperature (23.1 ± 5.7 °C). After cross-clamping of the distal ascending aorta, the remaining ascending aorta was resected, and the aortic root and valve were inspected and treated if necessary. Cardioplegia according to Brettschneider was given. Depending on the patient’s specific pathology, the operation was performed. While in selected cases it was possible to treat localized dissections with replacement of only the ascending aorta proximal to the cross clamp, most pathologies required more extensive surgery. After reaching the targeted temperature, the cross clamp was opened and the distal anastomosis of the hemiarch or arch prosthesis was performed in hypothermic circulatory arrest. Antegrade or retrograde selective cerebral perfusion was used as necessary.

After rewarming to normothermia and reperfusion, the patient was weaned from CPB. The surgical team ensured hemostasis and inserted chest tubes. After completion of the surgical procedure, the patient was transferred to the intensive care unit for further monitoring and treatment.

### 2.3. Statistical Analysis

Univariate comparisons of preoperative, operative, and postoperative variables were performed between women and men. Continuous and discrete variables were tested using Student’s t-test or the chi-square test, respectively. Long-term survival was analyzed using Kaplan–Meier curves and the log-rank test. All statistical tests were 2-sided, with the alpha level set at 0.05 for statistical significance and a 95% confidence interval (CI). All frequency data are presented as absolute numbers (percentages), continuous data as the mean ± standard deviation, and survival length as the median [95% CI].

The statistical remaining life expectancy at the time of presentation was calculated using data provided by the Federal Statistical Office of Germany based on the year of birth, age at presentation and sex. After matching between male and female patients according to the statistical remaining life expectancy (range ± 1 year), the resulting subgroup was compared using the same statistical tools.

The statistical computations were performed using Wizard Pro data analysis version 1.9.7 (Evan Miller, Chicago, IL, USA) and IBM SPSS Statistics 27 (IBM, Armonk, NY, USA). Matching between male and female patients was performed using the case control matching feature of IBM SPSS Statistics. Kaplan-Meier curves were created using GraphPad Prism version 10 (GraphPad Software, Boston, MA, USA).

## 3. Results

### 3.1. Patient Demographics and Cardiovascular Comorbidities

Over a period of 18 years, a total of 874 patients were operated on for acute type A aortic dissections. Of these, 313 patients (35.8%) were female, and 561 patients (64.2%) were male. The mean follow-up was 5.3 [5.0–5.7] years, with a 96% follow-up rate.

The mean age was 64.4 ± 13.6 years, the mean BMI (body mass index) was 27.3 ± 5.1 kg/m^2^ and the mean BSA (body surface area) was 2.0 ± 0.2 m^2^ (Table 1). DeBakey type I aortic dissection was diagnosed in 70.3% and DeBakey type II aortic dissection in 29.7% of cases. Relevant comorbidities were arterial hypertension (72.0%), smoking (21.2%), coronary artery disease (18.0%), diabetes mellitus (8.9%) and chronic obstructive pulmonary disease (COPD) (8.7%).

At the time of presentation, women were significantly older compared to men (69.1 ± 13.0 vs. 61.8 ± 13.3 years, *p* < 0.001) and had a significantly shorter remaining statistical life expectancy (18.6 ± 10.8 vs. 21.4 ± 10.4 years, *p* < 0.001). The women’s BMI and BSA were significantly lower compared to those of men (26.5 ± 5.3 vs. 27.8 ± 4.9 kg/m^2^ and 1.8 ± 0.2 m^2^ vs. 2.1 ± 0.2, each *p* < 0.001). Significantly more DeBakey type II aortic dissections were found in women (37.1% vs. 25.7%, *p* < 0.001). The incidence of systemic hypertension, diabetes mellitus and coronary artery disease (CAD) was similar in women and men (Table 1). Interestingly, the female patients smoked less often compared to the male patients (16.9% vs. 23.5%, *p* = 0.022), but showed a higher incidence of COPD (11.5% vs. 7.1%, *p* = 0.028).

### 3.2. Patient Preoperative Status

At the time of presentation, 8.7% of patients had required cardiopulmonary resuscitation and 23.8% had experienced circulatory shock. A bicuspid aortic valve was found in 4.2% of patients and an aortic valve insufficiency was diagnosed in 69.8% of patients. A total of 41.8% of patients presented with any malperfusion, including 13.5% of all patients with coronary malperfusion (Table 2).

The Penn classification was used to evaluate the degree of circulatory compromise at presentation. A total of 48.9% suffered from neither generalized (shock) nor localized malperfusion (Penn Aa). While 27.3% of patients experienced only localized malperfusion affecting one or several organ systems (Penn Ab), 9.4% had no localized malperfusion but presented with circulatory shock (Penn Ac). A total of 14.4% suffered from both localized and general malperfusion (Penn Abc).

At the time of presentation, the incidence of cardiopulmonary resuscitation, shock and aortic valve regurgitation showed no significant differences between both sexes. Female patients experienced less malperfusion, especially coronary (8.6% vs. 16.2%, *p* = 0.002) and renal (7.0% vs. 11.8%, *p* = 0.026) malperfusion, while there were no significant differences in cerebral, spinal, mesenteric and peripheral malperfusion.

A bicuspid aortic valve (1.6% vs. 5.2%, *p* = 0.004) was found more frequently in men than in women (Table 2).

### 3.3. Procedural Details

Isolated proximal replacement of the ascending aorta was performed in 27.3%, replacement of the ascending aorta with hemiarch replacement in 56.1% and replacement of the ascending aorta with complete arch replacement in 13.7% of the patients. The cardiopulmonary bypass time was 185 ± 86 min and the cross-clamp time was 98 ± 46 min. Procedures and perfusion details, as well as concomitant cardiac procedures, are listed in Table 3.

While the hemiarch procedure was the most prevalent surgical technique in both sexes, there were significant differences between men and women concerning the type of aortic repair. The isolated replacement of the ascending aorta was less frequent in women compared to men (24.3% vs. 29.1%). More hemiarch procedures (63.3% vs. 52.0%, *p* < 0.001) and fewer complete arch replacements (8.6% vs. 16.6%, *p* < 0.001) were performed in women, resulting in shorter CPB (173 ± 75 vs. 191 ± 91 min, *p* = 0.002) and cross-clamp times for women (92 ± 39 vs. 102 ± 49 min, *p* < 0.001). The target temperatures (22.9 ± 5.8 vs. 23.2 ± 5.6 °C, *p* = 0.479) were not significantly different. Concomitant procedures, such as cardiopulmonary bypass grafting (CABG), root replacement or valve replacement or repair, were not performed more frequently in either women or men.

### 3.4. Outcome

While male patients had fewer neurologic deficits after surgery than before, female patients had more neurologic deficits after surgery than before, as well as more neurologic deficits than male patients (18.8% vs. 11.9%, *p* = 0.020) (Table 4).

The overall in-hospital mortality after surgically treated acute type A aortic dissection was 12.2% (*n* = 107). In women and men, the in-hospital mortality was similar (11.5% vs. 12.7%, *p* = 0.618), as shown in Table 5. Long-term survival analysis (Figure 1) showed a significantly shorter median survival for women compared to men (9.8 [8.1–11.5] vs. 15.1 [11.9–18.4] years, *p* = 0.011).

### 3.5. Matched Subgroup Analysis

The study cohort was matched with respect to the remaining statistical life expectancy, resulting in a subgroup of 618 patients (309 male patients, 309 female patients). The same analysis was performed again on this subgroup.

A comparison of demographic data in this matched subgroup showed that while female patients were still significantly older than male patients (69.6 ± 12.3 vs. 65.6 ± 13.2 years, *p* < 0.001), there was no significant difference regarding the remaining statistical life expectancy between the two sexes (18.2 ± 10.0 vs. 18.3 ± 10.0 years, *p* = 0.885).

Differences in the distribution regarding the DeBakey classification were less pronounced and no longer showed a significant difference (*p* = 0.175), while BMI and BSA differences were still present in the matched subgroup. While female patients still showed higher rates of COPD (11.7% vs. 8.1%, *p* = 0.138) and lower rates of nicotine abuse (16.8% vs. 19.7%, *p* = 0.349), these differences were no longer significant (Table 6).

Male patients presented more often with a bicuspid aortic valve (4.5% vs. 1.3%, *p* = 0.017) and aortic valve regurgitation (76.4% vs. 65.7%, *p* = 0.003). While male patients experienced true lumen collapse more often than female patients, the difference in this subgroup was not significant (21.4% vs. 16.5%, *p* = 0.124).

While malperfusion still tended to occur more often in male patients, the difference in this subgroup was less pronounced and only coronary malperfusion occurred significantly more often in male patients (13.3% vs. 8.1%, *p* = 0.037).

Penn classification showed a homogenous distribution between both sexes (*p* = 0.612), adding further confidence to the comparability of male and female patients in this subgroup (Table 7). Similar to the overall study cohort, there were no significant differences in preoperative neurological status in this subgroup (*p* = 0.385).

The surgical technique varied between male and female patients (*p* < 0.001). Comparable to the total study cohort, the hemiarch procedure was the most popular type of repair in both sexes, but was used more often in female patients (63.1% vs. 49.2%). While the isolated replacement of the ascending aorta was more often used in male patients (41.1% vs. 24.3%), the complete replacement of the aortic arch was slightly favored in female patients (8.7% vs. 7.1%) (Table 8).

This approximation resulted in similar CPB (172 ± 75 vs. 174 ± 98 min, *p* = 0.765) and cross-clamping times (92 ± 40 vs. 93 ± 50 min, *p* = 0.681) for female and male patients, respectively.

Concomitant procedures, including CABG, root replacement and aortic valve repair or replacement, were not performed more frequently in either sex.

Overall, the statistical difference in remaining life expectancy between the two sexes was successfully eliminated in this matched subgroup compared with the overall study cohort, and an approximation between the two sexes in terms of hemodynamic status at presentation and surgical treatment performed was achieved.

Like in the full study cohort analysis, short-term survival (intraoperative, in-hospital and 30-day mortality) showed no significant differences between men and women. The median survival analyzed using the Kaplan–Meier estimator revealed that while survival in female patients still tended to be shorter, the difference was no longer statistically significant (9.8 [7.9–11.6] vs. 12.4 [10.1–14.7] years, *p* = 0.487). Furthermore, the incidence of postoperative neurological deficits also showed no significant differences between male and female patients (13.3% vs. 18.8%, *p* = 0.176) (Table 9 and Table 10, Figure 2).

## 4. Discussion

We present our data on the impact of sex on short- and long-term outcomes after surgically treated acute type A aortic dissection. It is important to understand sex-related differences in patients undergoing surgical repair of AAD because surgical treatment is the standard for AAD and data on sex differences, especially concerning long-term outcomes, are limited. To our knowledge, this represents the largest single-center study investigating sex differences in long-term survival after surgical treatment for AAD.

Our study analyzed the demographics, comorbidities, operative details and postoperative course of 874 patients with surgically treated AAD over a period of 18 years with respect to sex.

The mean age was about 65 years and women accounted for 1/3 of the entire study collective. Female patients were significantly older and smaller, and had a lower BMI, less history of smoking and more DeBakey type II aortic dissection compared to men. The remaining life expectancy of female patients was significantly shorter at the time of presentation. About 70% of patients suffered from arterial hypertension with no significant sex differences.

These demographic data and comorbidities are in line with data from the International Registry of Acute Aortic Dissection as well as other national and international registers [6,8,12,13,14,15,16,17]. The distribution regarding Penn classification roughly matches the distribution of other published studies, with slightly more Penn Ab and less Penn Ac patients in our observations [18,19,20]. Therefore, it should be possible to compare this study to other studies and to transfer our results to the general population.

It is interesting to note that while a history of smoking was more common in male patients, female patients were more likely to have COPD. This appears counter-intuitive and does not reflect the prevalence of the disease in Germany, but might be supported by some evidence that women are more susceptible to the development of COPD [21] as well as by the higher age of women in this study group.

Operational strategies varied between men and women and generally tended toward less extensive repair in female patients (more hemiarch and less total arch replacements), in line with the higher occurrence of the more limited DeBakey type II dissection in female patients.

Short-term survival (intraoperative mortality, in-hospital mortality and 30-day mortality) showed no significant differences. The overall in-hospital mortality of 12.2% was slightly lower compared to other international studies [14,17].

While the entire study cohort showed significantly shorter long-term survival for women, the matched subgroup analysis revealed that this can mostly be attributed to the already shorter remaining life expectancy of women at the time of presentation. The composition of the subgroup is more favorable for direct comparison, as evidenced by the similar hemodynamic status at presentation stratified by Penn classification, and the approximation of the employed surgical procedure. Furthermore, the differences in the neurologic outcome were also less pronounced in the matched subgroup. However, since women still tend to have shorter long-term survival, the missing statistical significance might potentially be explained by the smaller number of patients included in the subgroup analysis. Thus, further studies with a higher number of patients would be beneficial for more conclusive results regarding the impact of sex on long-term outcomes.

While several studies showed no significant differences between male and female patients regarding short- and long-term outcomes [8,9,10,16,22,23], other studies showed significant differences in in-hospital and 30-day mortality, to the detriment of women [6,7].

Nienaber et al. [7] evaluated sex differences in 1078 patients with acute type A dissections (32.1% female). Women were significantly older, had more hypertension and presented in a worse clinical condition compared to men. They found a significantly higher in-hospital mortality in women compared to men among patients with aortic type B dissections (30.1% vs. 21.0%). Surgically treated female patients with aortic type A dissections had a higher in-hospital mortality rate (31.9% vs. 21.9%, *p* = 0.013). They suggested that this was because of a later diagnosis and a more critical preoperative condition in women. In different age groups (age < 50 yo, 50–65 yo, 66-75 yo and >75 yo) they found a higher mortality rate for women between 66 and 75 years (36% vs. 16%, *p* < 0.001). However, due to the earlier study period (1996 to 2001), their results should not be directly compared to the present study.

Fukui et al. [9] showed in 504 patients operated on for AAD that female patients with AAD underwent complex surgery less frequently than male patients. DeBakey type II aortic dissection was more often found in women (18.4% vs. 9.7%, *p* = 0.070). Therefore, total arch replacement (19.2% vs. 44.8%; *p* < 0.001) and aortic root replacement (9.0% vs. 19.7%, *p* = 0010) were performed more often in men. Thus, the operation time (232 vs. 273 min, *p* < 0.001) and the aortic cross-clamp time (91 vs. 118 min, *p* < 0.001) were shorter for female patients. They showed that concomitant procedures were almost similar in both groups.

These results are similar to the present study. In our cohort, more complex surgery was also performed less frequently in female patients (arch replacements 8.6% vs. 16.6%, *p* < 0.001), resulting in shorter cross-clamp (92 ± 39 vs. 102 ± 49 min, *p* < =0.001) and cardiopulmonary bypass (173 ± 75 vs. 191 ± 91 min, *p* = 0.002) times for women. This increased complexity of the performed operation can probably be attributed to the extension of the aortic dissection, since significantly more DeBakey type II aortic dissections were found in women (37.1% vs. 25.7%, *p* < 0.001). Concomitant procedures, such as CABG, root replacement or valve replacement, were likewise not performed more frequently in either women or men.

Sabashnikov et al. [10] evaluated the impact of sex on long-term outcomes after AAD using propensity score matching to account for different preoperative risk profiles. They included 142 patients in their analysis and showed that women and men with similar risk profiles have no significant differences regarding short- and long-term survival. Their study investigated a similar time period and their results concerning 30-day mortality and survival at 5 years yielded results comparable to those from our study collective. Furthermore, the lack of significant differences in long-term survival after the elimination of preoperative differences is also supported by our results.

While our subgroup analysis eliminated the significant differences in the long-term outcome, the survival of female patients still tended to be shorter. This could possibly be explained by histologic differences in the aortic wall tissue between the sexes [24]. Unfortunately, we did not perform any histological examination as we do not have a histological database dating back to 2004 for this retrospective study.

Another important aspect could be the hormonal influence of younger women who are not yet or are just in the menopause. The protective effects of estrogen on the cardiovascular system might in part be responsible for the higher age of women at the time of diagnosis.

### Limitations

The significance of our results is limited by the retrospective design of our study. Surgical strategies and used prostheses have changed over the course of the 18 years. With the introduction of the frozen elephant trunk (FET) technique and, more recently, the availability of several branched FET prostheses, arch replacement has been performed more often at our center in recent years. While this should affect both sexes equally, we did not evaluate these changes over time.

Furthermore, this study only included patients who survived the preoperative preparations, including imaging and transfer to our center (if applicable) and operating theater. Sex differences in preoperative mortality were not investigated.

The cause of death was not obtained for every patient and our analysis did not differentiate between aortic-related mortality and other-cause mortality. While unlikely, a possible difference in aortic-related survival would not have been identified using our methods. This could be the subject of further research.

## 5. Conclusions

In conclusion, there are sex differences in the outcome of surgically treated acute type A aortic dissection. Sex influences the incidence—male patients are twice as common as female patients—and age at the time of presentation—women are significantly older compared to men. The preoperative conditions and comorbidities are nearly similar in female and male patients. Women more frequently have DeBakey type II aortic dissection and less frequently have bicuspid aortic valves and true lumen collapse, yet they have the same in-hospital mortality and a significantly worse long-term outcome. However, this result can mostly be explained by the preoperative lower remaining statistical life expectancy of female patients.

## Figures and Tables

**Figure 1 medicina-60-00443-f001:**
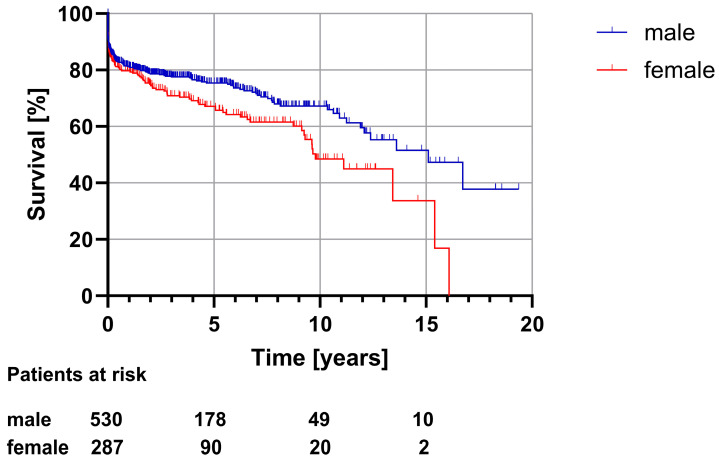
Long-term survival (total study cohort, *p* = 0.011).

**Figure 2 medicina-60-00443-f002:**
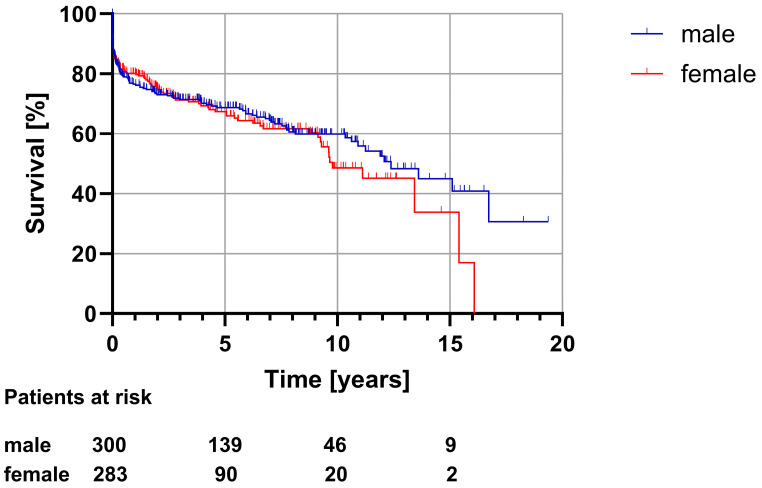
Long-term survival (matched subgroup, *p* = 0.487).

**Table 1 medicina-60-00443-t001:** Patient demographics (total study cohort).

Variable	Total (*n* = 874)	Male (*n* = 561)	Female (*n* = 313)	*p*-Value
DeBakey				**<0.001**
Type I	614 (70.3%)	417 (74.3%)	197 (62.9%)	
Type II	260 (29.7%)	144 (25.7%)	116 (37.1%)	
Demographics				
Age [years]	64.4 ± 13.6	61.8 ± 13.3	69.1 ± 13.0	**<0.001**
Remaining life expectancy [years]	20.4 ± 10.6	21.4 ± 10.4	18.6 ± 10.8	**<0.001**
BMI [kg/m^2^]	27.3 ± 5.1	27.8 ± 4.9	26.5 ± 5.3	**<0.001**
BSA [m^2^]	2.0 ± 0.2	2.1 ± 0.2	1.8 ± 0.2	**<0.001**
Comorbidities				
Arterial hypertension	629 (72.0%)	406 (72.4%)	223 (71.2%)	0.723
Diabetes mellitus	78 (8.9%)	52 (9.3%)	26 (8.3%)	0.632
Coronary artery disease	157 (18.0%)	102 (18.2%)	55 (17.6%)	0.822
Nicotine	185 (21.2%)	132 (23.5%)	53 (16.9%)	**0.022**
COPD	76 (8.7%)	40 (7.1%)	36 (11.5%)	**0.028**
Previous cardiac surgery	58 (6.6%)	40 (7.1%)	18 (5.8%)	0.432

BMI: body mass index; BSA: body surface area; COPD: chronic obstructive pulmonary disease.

**Table 2 medicina-60-00443-t002:** Clinical status at admission (total study cohort).

Variable	Total (*n* = 874)	Male (*n* = 561)	Female (*n* = 313)	*p*-Value
Intubated/ventilated	124 (14.2%)	80 (14.3%)	44 (14.1%)	0.934
Cardiopulmonary resuscitation	76 (8.7%)	49 (8.7%)	27 (8.6%)	0.957
Shock	208 (23.8%)	135 (24.1%)	73 (23.3%)	0.814
Bicuspid aortic valve	37 (4.2%)	32 (5.2%)	5 (1.6%)	**0.004**
Aortic valve regurgitation	610 (69.8%)	403 (71.8%)	207 (66.1%)	0.078
True lumen collapse	206 (23.6%)	155 (27.6%)	51 (16.3%)	**<0.001**
Pericardial effusion/tamponade				0.155
Pericardial effusion	379 (43.4%)	201 (41.0%)	149 (47.6%)	
Tamponade	145 (16.6%)	95 (16.9%)	50 (16.0%)	
Malperfusion	365 (41.8%)	259 (46.2%)	106 (33.9%)	**<0.001**
Coronary	118 (13.5%)	91 (16.2%)	27 (8.6%)	**0.002**
Cerebral	135 (15.4%)	90 (16.0%)	45 (14.4%)	0.514
Spinal	21 (2.4%)	14 (2.5%)	7 (2.2%)	0.810
Mesenteric	102 (11.7%)	73 (13.0%)	29 (9.3%)	0.098
Renal	88 (10.1%)	66 (11.8%)	22 (7.0%)	**0.026**
Peripheral	129 (14.8%)	90 (16.0%)	39 (12.5%)	0.152
Penn Classification				**0.005**
Aa	427 (48.9%)	255 (45.5%)	172 (55.0%)	
Ab	239 (27.3%)	171 (30.5%)	68 (21.7%)	
Ac	82 (9.4%)	47 (8.4%)	35 (11.2%)	
Abc	126 (14.4%)	88 (15.7%)	38 (12.1%)	
Neurological preoperative status				0.722
No neurologic deficits	385 (78.4%)	441 (78.6%)	244 (78.0%)	
Neurologic deficits	139 (15.9%)	86 (15.3%)	53 (16.9%)	
Not obtainable	50 (5.7%)	34 (6.1%)	16 (5.1%)	

**Table 3 medicina-60-00443-t003:** Procedural details (total study cohort).

Variable	Total (*n* = 874)	Male (*n* = 561)	Female (*n* = 313)	*p*-Value
Aortic replacement				**<0.001**
Ascending	239 (27.3%)	163 (29.1%)	76 (24.3%)	
Ascending, hemiarch	490 (56.1%)	292 (52.0%)	198 (63.3%)	
Ascending, arch	120 (13.7%)	93 (16.6%)	27 (8.6%)	
Other proximal repair	25 (2.9%)	13 (2.3%)	12 (3.8%)	
Perfusion details				
Cardiopulmonary bypass time [min]	185 ± 86	191 ± 91	173 ± 75	**0.002**
Cross-clamp time [min]	98 ± 46	102 ± 49	92 ± 39	**<0.001**
Lowest temperature [°C]	23.1 ± 5.7	23.2 ± 5.6	22.9 ± 5.8	0.479
Concomitant cardiac procedures				
Coronary artery bypass grafting	118 (13.5%)	79 (14.1%)	39 (12.5%)	0.501
Root replacement	69 (7.9%)	48 (8.6%)	21 (6.7%)	0.332
Aortic valve				0.254
Repair	624 (71.4%)	400 (71.3%)	224 (71.6%)	
Replacement	87 (10.0%)	62 (11.1%)	25 (8.0%)	

**Table 4 medicina-60-00443-t004:** Postoperative course (total study cohort).

Variable	Total (*n* = 874)	Male (*n* = 561)	Female (*n* = 313)	*p*-Value
Tracheotomy	56 (6.4%)	38 (6.8%)	18 (5.8%)	0.554
Rethoracotomy	101 (11.6%)	72 (12.8%)	29 (9.3%)	0.114
Renal failure requiring dialysis				0.107
Temporary	131 (15.0%)	93 (16.6%)	38 (12.1%)	
Permanent	9 (1.0%)	4 (0.7%)	5 (1.6%)	
Neurological postoperative status				**0.020**
No neurologic deficits	680 (77.8%)	450 (80.2%)	230 (73.5%)	
Neurologic deficits	126 (14.4%)	67 (11.9%)	59 (18.8%)	
Not obtainable	68 (7.8%)	4.45 (7.8%)	24 (7.7%)	

**Table 5 medicina-60-00443-t005:** Short- and long-term survival (total study cohort).

Variable	Total (*n* = 874)	Male (*n* = 561)	Female (*n* = 313)	*p*-Value
Intraoperative mortality	23 (2.6%)	15 (2.7%)	8 (2.6%)	0.917
In-hospital mortality	107 (12.2%)	71 (12.7%)	36 (11.5%)	0.618
30-day mortality (*n* = 842) ^1^	112 (13.3%)	69 (12.7%)	43 (14.5%)	0.458
Long-term survival				
Median survival [years]	13.4 [10.8–16.0]	15.1 [11.9–18.4]	9.8 [8.1–11.5]	**0.011**
5-year survival	72.3%	75.0%	66.4%	
10-year survival	59.4%	65.9%	45.0%	
15-year survival	42.6%	47.3%	16.9%	

^1^ Only patients with completed 30-day follow-up were included.

**Table 6 medicina-60-00443-t006:** Patient demographics (matched subgroup).

Variable	Total (*n* = 618)	Male (*n* = 309)	Female (*n* = 309)	*p*-Value
DeBakey				0.175
Type I	406 (65.7%)	211 (68.3%)	195 (63.1%)	
Type II	212 (34.3%)	98 (31.7%)	114 (36.9%)	
Demographics				
Age [years]	67.6 ± 12.9	65.6 ± 13.2	69.6 ± 12.3	**<0.001**
Remaining life expectancy [years]	18.2 ± 10.0	18.3 ± 10.0	18.2 ± 10.0	0.885
BMI [kg/m^2^]	27.0 ± 4.9	27.5 ± 4.4	26.5 ± 5.3	**0.012**
BSA [m^2^]	1.9 ± 0.2	2.1 ± 0.2	1.8 ± 0.2	**<0.001**
Comorbidities				
Arterial hypertension	457 (73.9%)	236 (76.4%)	213 (71.5%)	0.169
Diabetes mellitus	60 (9.7%)	34 (11.0%)	26 (8.4%)	0.277
Coronary artery disease	121 (19.6%)	66 (21.4%)	55 (17.8%)	0.265
Nicotine	113 (18.3%)	61 (19.7%)	52 (16.8%)	0.349
COPD	61 (9.9%)	25 (8.1%)	36 (11.7%)	0.138
Previous cardiac surgery	41 (6.6%)	24 (7.8%)	17 (5.5%)	0.258

BMI: body mass index; BSA: body surface area; COPD: chronic obstructive pulmonary disease.

**Table 7 medicina-60-00443-t007:** Clinical status at admission (matched subgroup).

Variable	Total (*n* = 618)	Male (*n* = 309)	Female (*n* = 309)	*p*-Value
Intubated/ventilated	92 (14.9%)	48 (15.5%)	44 (14.1%)	0.651
Cardiopulmonary resuscitation	53 (8.6%)	26 (8.4%)	27 (8.7%)	0.886
Shock	154 (24.9%)	82 (26.5%)	72 (23.3%)	0.352
Bicuspid aortic valve	18 (2.9%)	14 (4.5%)	4 (1.3%)	**0.017**
Aortic valve regurgitation	439 (71.0%)	236 (76.4%)	203 (65.7%)	**0.003**
True lumen collapse	117 (18.9%)	66 (21.4%)	51 (16.5%)	0.124
Pericardial effusion/tamponade				0.183
Pericardial effusion	271 (43.9%)	125 (40.5%)	146 (47.2%)	
Tamponade	113 (18.3%)	63 (20.4%)	50 (16.2%)	
Malperfusion	220 (35.6%)	116 (37.5%)	104 (33.7%)	0.313
Coronary	66 (10.7%)	41 (13.3%)	25 (8.1%)	**0.037**
Cerebral	89 (14.4%)	44 (14.2%)	45 (14.6%)	0.909
Spinal	14 (2.3%)	7 (2.3%)	7 (2.3%)	1.000
Mesenteric	60 (9.7%)	31 (10.0%)	29 (9.4%)	0.786
Renal	53 (8.6%)	31 (10.0%)	22 (7.1%)	0.196
Peripheral	79 (12.8%)	40 (12.9%)	39 (12.6%)	0.904
Penn Classification				0.612
Aa	329 (53.2%)	159 (51.5%)	170 (55.0%)	
Ab	135 (21.8%)	68 (22.0%)	67 (21.7%)	
Ac	69 (11.2%)	34 (11.0%)	35 (11.3%)	
Abc	85 (13.8%)	48 (15.5%)	37 (12.0%)	
Neurological preoperative status				0.385
No neurologic deficits	495 (80.1%)	254 (82.2%)	241 (78.0%)	
Neurologic deficits	94 (15.2%)	41 (13.3%)	53 (17.2%)	
Not obtainable	29 (4.7%)	14 (4.5%)	15 (4.9%)	

**Table 8 medicina-60-00443-t008:** Procedural details (matched subgroup).

Variable	Total (*n* = 618)	Male (*n* = 309)	Female (*n* = 309)	*p*-Value
Aortic replacement				**<0.001**
Ascending	202 (32.7%)	127 (41.1%)	75 (24.3%)	
Ascending, hemiarch	347 (56.1%)	152 (49.2%)	198 (63.1%)	
Ascending, arch	49 (7.9%)	22 (7.1%)	27 (8.7%)	
Other proximal repair	20 (3.2%)	8 (2.6%)	12 (3.9%)	
Perfusion details				
Cardiopulmonary bypass time [min]	173 ± 87	174 ± 98	172 ± 75	0.765
Cross-clamp time [min]	92 ± 45	93 ± 50	92 ± 40	0.681
Lowest temperature [°C]	23.4 ± 6.0	23.9 ± 6.2	23.0 ± 5.7	0.060
Concomitant cardiac procedures				
Coronary artery bypass grafting	85 (13.8%)	47 (15.2%)	38 (12.3%)	0.293
Root replacement	36 (5.8%)	15 (4.9%)	21 (6.8%)	0.303
Aortic valve				0.079
Repair	461 (74.6%)	241 (78.0%)	220 (71.2%)	
Replacement	50 (8.1%)	25 (8.1%)	25 (8.1%)	

**Table 9 medicina-60-00443-t009:** Postoperative course (matched subgroup).

Variable	Total (*n* = 618)	Male (*n* = 309)	Female (*n* = 309)	*p*-Value
Tracheotomy	43 (7.0%)	25 (8.1%)	18 (5.8%)	0.268
Rethoracotomy	68 (11.0%)	41 (13.3%)	27 (8.7%)	0.072
Renal failure requiring dialysis				**0.038**
Temporary	95 (15.4%)	58 (18.8%)	37 (12.0%)	
Permanent	7 (1.1%)	2 (0.6%)	5 (1.6%)	
Neurological postoperative status				0.176
No neurologic deficits	471 (76.2%)	243 (78.6%)	228 (73.8%)	
Neurologic deficits	99 (16.0%)	41 (13.3%)	58 (18.8%)	
Not obtainable	48 (7.8%)	25 (8.1%)	23 (7.4%)	

**Table 10 medicina-60-00443-t010:** Short- and long-term survival (matched subgroup).

Variable	Total (*n* = 618)	Male (*n* = 309)	Female (*n* = 309)	*p*-Value
Intraoperative mortality	15 (2.4%)	8 (2.6%)	7 (2.3%)	0.794
In-hospital mortality	79 (12.8%)	44 (14.2%)	35 (11.3%)	0.278
30-day mortality (*n* = 597) ^1^	84 (14.1%)	42 (13.8%)	42 (14.3%)	0.855
Long-term survival				
Median survival [years]	11.9 [9.8–14.1]	12.4 [10.1–14.7]	9.8 [7.9–11.6]	0.487
5-year survival	68.0%	68.2%	66.6%	
10-year survival	54.8%	58.7%	45.2%	
15-year survival	38.5%	40.8%	16.9%	

^1^ Only patients with completed 30-day follow-up were included.

## Data Availability

The datasets presented in this article are not readily available because of privacy restrictions. Reasonable requests to access the datasets should be directed to the corresponding author.

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
