# Peer review of "Sex Differences in Short- and Long-Term Survival after Acute Type A Aortic Dissection"

_medicina, 2024, doi:10.3390/medicina60030443_

Round 1

Reviewer 1 Report

Comments and Suggestions for Authors

This study examined sex differences in the treatment and outcomes of Acute Type A aortic dissection (AAD) among 874 patients (313 women, 561 men) from 2004 to 2021. It found that women were older with a shorter remaining life expectancy at the time of AAD diagnosis and more frequently had DeBakey Type II dissections. Despite similar in-hospital mortality rates, women initially showed a significantly shorter long-term survival compared to men. However, this difference was not observed when comparing groups with similar life expectancies, suggesting that the apparent worse outcome for women could largely be attributed to their shorter life expectancy at the time of AAD diagnosis.

General comments

This is a manuscript addressing a topic “Sex Differences in the short- and long-term Survival after Acute Type A Aortic Dissection”. However, some concerns need to be addressed.

Specific comments

Major

1)    Line 143: The matching between male and female patients was according to the statistical remaining life expectancy (range ± 1 year). The rational of the range setting would be vague; the average life expectancy was adapted to the female group (18 years) but not to male. The use of longer life expectancy cohort (e.g. 21 years) would have different results from this study. Because this is an important study addressing the sex difference, the reviewer recommends the additional analysis including the other range of life expectancy, e.g., group A: 16-18, B: 18-20, C: 20-22 years. This kind of analysis would provide more impressive results in this field.

Minor

1)    Table 4, 7, 9: the words “without finding” and “pathologic” would be revised: e.g. symptomatic without CT findings and symptomatic.

2)    The figures of Kaplan–Meier curves would need numbers at risk.

Author Response

We thank the reviewer for their review and detailed suggestions on how to improve this manuscript.

Major:

1) It is correct that in the subgroup analysis, the remaining life expectancy was adjusted mostly to the female study group because we used 1-on-1 matching and the female study cohort was smaller. This resulted in almost all female patients being included in the subgroup, while only about 55% of males were included.

To address this issue, we split the matched subgroup into further subgroups based on the remaining life expectancy. Using the suggested groups A-C, we did not find any significant differences in the long-term survival. Due to the relatively low number of patients in these groups, we also tried this with larger groups (remaining life expectancy 0-10 years, 10-20 years, etc.) and again found no significant differences in the long-term survival. We also used all of these subgroups on the whole (unmatched) study cohort and again found no significantly different long-term survival. We thank the reviewer for their suggestion and believe this analysis confirms our manuscripts results and conclusions. In our opinion, the comparison of subgroups based on the remaining life expectancy is another possible option to compare male and female patients, resulting in the same conclusions.

After careful consideration, we believe that the proposed subgroup analysis is beyond the scope of this manuscript. It further reduces the sample size and thus decreases the power. We opted for the matching and were able to demonstrate that the matched subgroup showed an approximation in almost all fields (remaining life expectancy, hemodynamic status preoperatively, type of surgical treatment, and long-term survival). We therefore conclude that matching based on the remaining life expectancy as performed is suitable for our analysis, and we do not see how the proposed

If your opinion differs we kindly ask to note in the review of this revision and we will be happy to add a subgroup analysis based on groups with a similar remaining life expectancy as a second way to achieve comparable groups.

Minor:

1) We have changed these to "(no) neurologic deficits", as the item was obtained using primarily the clinical examination pre- and postoperatively.

2) We have added the number of patients at risk to figures 1 and 2.

Reviewer 2 Report

Comments and Suggestions for Authors

I would like to thank the editor for allowing to review this manuscript.

I would also like to congratulate the authors on their work. I have a few questions.

1. Line 60 to 62 should be in the results section.

2. The number of included patients, percentage of female patients and follow-up data, etc (line 67 to 93) should be in the results section. In the methods section it should be described how the patients were selected (consecutive patients?), whether the study was approved by an ethics committee and which patients were excluded.

3. A type error has been made in line 181 (“shower”)

4. How was the matched group determined, which statistics program was used? This should be stated in the methods section.

5. p-value should be added to the text in line 216, line 217 and line 219.

6. I think that an extra survival graph should be added regarding “dissection-related” survival. This could further confirm the conclusions of the authors that long-term survival of female patients was not different when compared to male patients after type A dissection. This could explain why female patients with more DeBakey Type II AD still have a worse mortality.

7. “Surgical” should be “Surgically” in line 295

8. Was the FET prostheses also used during the study period? Could this have any effect on the outcomes during these study years (inclusion of more complex anatomy?).

Author Response

Thank you for thoroughly reading this manuscript and your valuable feedback.

1. We have moved this to the results section.

2. We agree with the reviewer and have rearranged the manuscript to only include number of patients etc. in the results section. The ethics statement from the end of the article has also been included in the methods section as suggested.

3. We have corrected this typo.

4. We used SPSS for the matching, we have updated and rearranged the statistical analysis section to include this information.

5. Since this is the same statistical comparison, we have added the p-value in the preceding line.

6. This is a very interesting point and should be the subject of further research. However, for some patients we were only able to obtain the date of death from government resources (resident registration offices) and therefore do not have information about whether the cause of death is dissection related for every patient. We think the conclusions of this manuscript are still valid without this analysis, but we have added the missing differentiation between aortic and non aortic death in the limitations section.

7. We have corrected this typo.

8. The FET prosthesis was used in most arch replacement surgeries and arch replacement using FET has been performed more often in recent years. While the  use of FET itself might of course have an impact on survival, this should affect both sexes. We have clarified this in the limitations section. Patients were treated surgically regardless of the complexity of their anatomy, only the type of repair (e. g. arch replacement using FET) differed between anatomical conditions, time periods, surgeons, and available prostheses. I hope this answers your question.

Round 2

Reviewer 1 Report

Comments and Suggestions for Authors

The authors have corrected the manuscript according to the reviewer's comments and explained the matching of groups by providing additional information. 

Reviewer 2 Report

Comments and Suggestions for Authors

i advise to accept the manuscript in current form